# Process Development and Synthesis of Process-Related Impurities of an Efficient Scale-Up Preparation of 5,2′-Dibromo-2,4′,5′-Trihydroxy Diphenylmethanone as a New Acute Pyelonephritis Candidate Drug

**DOI:** 10.3390/molecules25030468

**Published:** 2020-01-22

**Authors:** Xiu E Feng, Ke Meng Cui, Qing Shan Li, Zi Cheng Wu, Fei Lei

**Affiliations:** 1School of Pharmaceutical Science, Shanxi Medical University, 56 Xinjian South Road, Taiyuan 030001, China; cuikem913@163.com; 2Shanxi Key laboratory of Chronic Inflammatory Targeted Drugs, School of Chinese Materia Medica, Shanxi University of Traditional Chinese Medicine, 121 University Street, Jinzhong 030602, China; 3Beijing Zhendong Guangming Pharmaceutical Research Institute, 18 Chuangye Road, Beijing 100089, China; wuzicheng@zdjt.com (Z.C.W.); leifei@zdjt.com (F.L.)

**Keywords:** 5,2′-dibromo-2,4′,5′-trihydroxydiphenylmethanone, process development, process-related impurity, acute pyelonephritis, polymorph

## Abstract

Based on a foregoing gram-scale laboratory process, an efficient scale-up preparation process of 5,2′-dibromo-2,4′,5′-trihydroxydiphenylmethanone (**LM49-API**), a new acute pyelonephritis candidate drug, was developed and validated aiming to reduce by-products and achieve better impurity profiles. Meanwhile, the polymorph of **LM49-API** and process-related impurities were also investigated. Ultimately, the optimal reaction conditions were verified by evaluating the impurity profiles and their formation during the synthesis. Six process-related impurities were synthesized and identified, being useful for the quality control of **LM49-API**. Its finalized preparation process was further validated at 329–410 g scale-up production in 53.4–57.1% overall yield with 99.95–99.98% high-performance liquid chromatography (HPLC) purity, and it is currently viable for commercial production. **LM49-API-imC** and **LM49-API-imX** were identified as the main single impurities in **LM49-API**, with the content controlled to be less than 0.03%.

## 1. Introduction

Acute pyelonephritis (APN) is a bacterial infection of the renal pelvis, accompanied by severe symptoms that range from mild discomfort to life-threatening illness or death [1]. If given prompt and adequate treatment, APN can be completely cured, and no complications occur. However, delayed or inappropriate treatment of APN may lead to the development of seriously repeated attacks and life-threatening infection, which can finally cause renal scarring and impairment of kidney function [2]. At present, the appropriate antimicrobial drugs remain the main selection for the clinical treatment of APN [2,3]. However, due to the misuse of antimicrobial drugs, drug-resistant strains continuously appear, and APN is easy to relapse after treatment, which affects the clinically curative effects thereof [4]. Therefore, it is imperative to develop novel treatment strategies to improve therapeutic options.

5,2′-Dibromo-2,4′,5′-trihydroxydiphenylmethanone (**LM49-API**) (shown in Scheme 1), a polyphenolic compound developed by our research group, exhibits strong inhibitory inflammation and antioxidant stress abilities [5,6,7,8]. Recently, we reported its therapeutic effects in an APN rat model. It is interesting that **LM49-API** does not directly inhibit bacteria; rather, its therapeutic efficacy on APN is closely related to inhibiting inflammation and regulating immune responses of T lymphocytes [9,10]. These continuous research results suggest its prospectively important clinical application in the treatment of APN. In order to explore the preclinical research and development of the new drug, it is necessary to develop an active pharmaceutical ingredients (API) scale-up preparation process suitable for commercial production, conduct the polymorphism analysis, investigate impurity profiling, and control impurity limits. In this way, we could obtain uniform quality, safe, and reliable API sample for druggability research.

The original laboratory preparation process of candidate compound **LM49-API** is at a gram scale [5], which is not conducive for its commercial production with some emerged drawbacks upon scale-up production: specifically, (i) difficult product purification in Friedel–Crafts acylation step; (ii) use of hazardous reagents, such as liquid bromine in bromination reaction step; (iii) release of a large amount of heat during neutralization with aqueous ammonia after the ending of bromination; (iv) difficult removal of by-products in bromination step; (v) severe temperature requirement of −78 °C in demethylation step; (vi) use of class 2 solvent methanol (MeOH) during recrystallization of **LM49-API**; (vii) lower overall yield of 47.0%. To address these problems, we focused on the improvement of the **LM49-API** preparation process aiming to develop an efficient, practical, economic, scalable, and safely reproducible process with mild reacting conditions, controllable impurities, easy purification, and high yield for its commercial production. In parallel, the potential process-related impurities were investigated and speculated for their formation pathway according to the organic reaction mechanism, and fully synthesized by chemical methods. Thus, the impurity reference substances were obtained, and the related impurities in the test samples were analyzed, identified, and controlled by high-performance liquid chromatography (HPLC).

## 2. Results and Discussion

### 2.1. Process Optimization

The main purpose of preclinical API process development is to prepare sufficient API samples so as to rapidly conduct the safety evaluation and collect data on safety, drug metabolism, and biological activity [11]. The original gram-scale laboratory preparation process of **LM49-API** is as follows [5]: (i) 2-methoxybenzoic acid reacted with anhydrous thionyl chloride (SOCl_2_) in the presence of *N,N*-dimethyl formamide (DMF) to prepare corresponding acyl chloride, then conducted the Friedel–Crafts acylation reaction with 1,2-dimethoxybenzene (**SM01**) in anhydrous dichloromethane (CH_2_Cl_2_) under the catalysis of aluminum chloride (AlCl_3_) to obtain the intermediate **LM49-01** by column chromatography separation. (ii) **LM49-01** was brominated with liquid bromine in acetic acid (HOAc), then neutralized with aqueous ammonia in the post-treatment, and separated by column chromatography to gain the key intermediate **LM49-02**. (iii) **LM49-02** was demethylated with boron tribromide (BBr_3_) in anhydrous CH_2_Cl_2_, and the crude product was recrystallized with methanol to obtain the target product **LM49-API**. In order to meet the large-scale commercial production of **LM49-API**, the reagent ratio, reaction temperature, reaction time, bromination reagent, and the purification conditions of intermediates and the final product were optimized and improved in this paper. The preparation process of **LM49-API** was shortened to three steps from the original four steps, achieving a high yield, easily available reagents, low cost, and simple separation and purification [5] (Scheme 1).

#### 2.1.1. Friedel–Crafts Acylation Step

SOCl_2_ was used in the original process. When exposed to water, it releases irritating toxic and harmful gases, such as sulfur dioxide and hydrogen chloride. In this paper, purchased 2-methoxybenzoyl chloride (**SM02**) directly reacted with **SM01** to prepare **LM49-01** by Friedel–Crafts acylation, thus avoiding the use of SOCl_2_. In the experiment, it was found that: (i) Since AlCl_3_ is insoluble in CH_2_Cl_2_, a small amount of solvent (**SM01**/CH_2_Cl_2_ was 1 g/5 mL) will lead to a thick reaction system, but increasing the CH_2_Cl_2_ amount will lead to rising costs. Finally, the reagent ratio of **SM01** to CH_2_Cl_2_ was chosen to be 1 g/10 mL. (ii) When the molar ratio of **SM01** to **SM02** was 1/1, **SM01** was found to be of incomplete reaction, being difficult to be removed as an organic impurity. **SM02** is easily hydrolyzed to 2-methoxybenzoic acid. If excess **SM02** was used in this step, it could be quickly removed by washing with 5% NaOH aqueous solution. The molar ratio of **SM01** to **SM02** was then determined to be 1/1.05. (iii) When the reaction temperature was kept between 0 °C and 30 °C, all reagents completely reacted with a higher yield. Meanwhile, we found that increasing the reaction temperature to 30 ± 5 °C will result in more impurities and a lower yield. In consideration of decreasing impurities and cost, the reaction temperature was preferably 15 ± 5 °C. (iv) The reaction process was relatively rapid and could be completely transformed within 1 h. If the reaction time was prolonged, more impurities were found. The reaction time was finally determined as 1 h. (v) In the presence of AlCl_3_, increasing the temperature resulted in demethylation of the product. Hence, the reaction temperature and the amount of AlCl_3_ should be controlled. When the molar ratio of **SM01** to **AlCl_3_** was 1/1, the reaction was completed with the highest yield (Table 1).

In the original preparation process, **LM49-01** was separated and purified by column chromatography. In this paper, the crude product of **LM49-01** was directly purified using the mixed solvents of CH_2_Cl_2_/*n*-hexane or ethyl acetate/*n*-hexane. When the CH_2_Cl_2_/*n*-hexane system was employed, its different proportions were screened. When the reagent ratio of **LM49-01**/CH_2_Cl_2_/*n*-hexane was 1 g/1 mL/20 mL, a white crystal powder product was obtained with a higher yield and purity. However, when changing the ratio to 1 g/1 mL/30 mL, the yield and purity were significantly reduced. A white block solid product was obtained, which may have contained more impurities. We also found that when the ratio was changed to 1 g/2 mL/40 mL, a lower yield was given due to the excessive use of solvent. In addition, when an ethyl acetate/*n*-hexane system was used for purification, various ratios were tried, and the yield was lower than 50%. Therefore, the reagent ratio of **LM49-01**/CH_2_Cl_2_/*n*-hexane for crystallization was determined as 1 g/1 mL/20 mL (Table 2).

The finally determined process of this step exhibits more advantages over the prior technology. For one thing, the reaction time was shortened significantly, and the use of the dangerous reagent thionyl chloride was prevented. For another, purification of **LM49-01** was easy by stirring to crystallize product at 20 ± 5 °C. This process is more suitable for scale-up production.

#### 2.1.2. Bromination Reaction Step

Liquid bromine is volatile, highly toxic, and dangerous for post-treatment. In order to avoid the use of liquid bromine, we mainly investigated several solid bromination reagents, such as *N*-bromosuccinimide (NBS), dibromohydantoin (DBDMH), and pyridine tribromide (PyBr_3_) [12,13,14]. The results in Table 3 present that when NBS and DBDMH were used as bromination reagents in tetrahydrofuran (THF), the reactions were completed in 5 h and 24 h, respectively, by stirring at room temperature. However, for PyBr_3_ bromination in MeOH, the reaction was completed with refluxing. By comparison, DBDMH and PyBr_3_ are more expensive than NBS, the required reaction time is longer, and heating is required in the case of PyBr_3_. Moreover, in consideration of the large-scale preparation in prospectively commercial production, the use of NBS as a bromination reagent displays more advantages including a shorter reaction time, low cost, low energy consumption, and simple post-treatment. Therefore, NBS was selected as a final bromination reagent in this step. Next, we mainly explored the influence of solvent, reagent ratio, reaction temperature, reaction time, and recrystallization conditions. As shown in Table 3, (i) when MeOH, HOAc, DMF, and CH_2_Cl_2_ were used as reaction media, thin-layer chromatography (TLC) monitoring indicated that the reaction was incomplete with obvious impurities in the lower yield. However, when THF was applied as the medium, the yield was significantly increased, and THF was therefore selected as the reaction solvent. (iii) With the decrease of solvent dosage, NBS dosage was also greatly reduced. At a determined **LM49-01**/THF ratio of 1 g/5 mL, the amount of NBS was screened. The results reveal that when the molar ratio of **LM49-01** to NBS was 1/4, the reaction was complete in 5 h with the highest yield. (3) The reaction temperature was screened between 0–65 °C. At 20 ± 5 °C, the reaction can be completed within 5 h with the highest yield. With increasing or decreasing temperature, the yield was significantly decreased. Especially when refluxing was performed for 24 h, a large amount of mono-brominated impurity **LM49-02-imB** (depicted in Scheme 2) was found.

The results in Table 4 show that for the same type of solvent and recrystallization method, 15 °C was more conducive to the precipitation of the product than 25 °C with a slightly higher yield but lower purity. Hence, we chose 25 °C as the crystallization temperature. Under the condition of same temperature and solvent, the yield and purity of stirring crystallization were higher than that of standing crystallization, with the former finally adopted. We also compared the use of methanol and ethanol as recrystallization solvents. No significant difference was found in yield and purity. In order to reduce the use of class 2 solvents, ethanol was preferred as the recrystallization solvent, and the amount of ethanol was further selected. With the increase of the ethanol amount, the yield of the product decreased, and the purity did not change considerably. Finally, the ratio of **LM49-02** to ethanol was selected as 1 g/10 mL.

Compared with the original technology, the finally determined bromination process presents the following advantages: (i) NBS was used to replace liquid bromine, which is convenient for feeding, the needed reagents are readily available, and it is suitable for scale-up production. (ii) The solvent was changed to THF. The post-treatment has therefore become easier. After the reaction was finished, the mixture was poured into ice water to stir and precipitate granular solid, and the crude product was obtained through filtration. (iii) In the purification process, the qualified product can be obtained by single recrystallization with ethanol.

#### 2.1.3. Demethylation Reaction Step

Using BBr_3_ as the demethylated reagent, we mainly investigated the influence of **LM49-02**/BBr_3_ ratio, feed temperature, reaction time, and recrystallization conditions. The research results in Table 5 exhibit that when the molar ratio of **LM49-02** to BBr_3_ was 1/3, the reaction could be completed within 2 h. When decreasing the amount of BBr_3_ with a **LM49-02**/BBr_3_ ratio of 1/2, more impurities were detected by TLC. When feeding at 15 ± 5 °C, the temperature will rise sharply, which may lead to the spillage of reaction mixture. When feeding at 0 ± 5 °C, the temperature can be maintained below 10 °C. However, considering the prospectively larger scale production, the overall volume of the reaction was increased, and the specific surface area was then decreased, resulting in slower heat dissipation rate, and there might be a risk. Therefore, the reaction temperature was further lowered. It was found that when the temperature of the system was always kept below 0 °C and when feeding BBr_3_ at −15 ± 5 °C, the reaction was stable and safe. Finally, BBr_3_ was added at −15 ± 5 °C.

Different recrystallization solvents were selected. The results in Table 6 reveal that when methanol/water and ethanol/water were used as recrystallized solvents, the yield and purity were higher than those of other solvents. Considering that methanol is a class 2 solvent, ethanol/water was preferred as the crystallization solvent. When the **LM49-API**/ethanol/water ratio was 3 g/12 mL/6 mL, the product was obtained with the lowest impurity content. However, in the crystallization process, due to the low solvent consumption, the system was thick and difficult to stir, and the solid was likely to stick to the wall of the container. Therefore, increasing the amount of solvent to a **LM49-API**/ethanol/water ratio of 3 g/12 mL/12 mL, the product was easy to precipitate and be separated at the highest yield.

Ultimately, the feeding temperature of BBr_3_ in demethylation step was improved from −78 °C to −15 ± 5 °C compared with original process, thus, reducing energy consumption. Moreover, in the recrystallization of **LM49-API**, the **LM49-API**/ethanol/water ratio was determined as 1 g/4 mL/4 mL, avoiding the use of the class 2 solvent methanol.

### 2.2. Process Verification

In the improved laboratory process, the one-time feeding scale of **SM01** was 5 g. According to the requirements of new drug development, the final process needs to be verified through step-by-step amplification. In order to further confirm the stability and feasibility of the process parameters, the production scale of API was increased by a factor of 60, that is, increasing the one-time feeding amount of **SM01** to 300 g. The preparation process of **LM49-API** was verified by three batches of production.

#### 2.2.1. Process Verification of Friedel–Crafts Acylation Step

As presented in Table 7 and Table 8, three batches of **LM49-01** samples were obtained at a scale of 461–490 g. In view of 2-methoxyphenol (**SM01-imA**) from **SM01** being a potential impurity and **SM02** being easily hydrolyzed to 2-methoxybenzoic acid (**SM02-imA**), the control substances of **SM01**, **SM02-imA**, **SM01-imA**, **LM49-01-imA**, and **LM49-01**, as well as their mixture and **LM49-01** test sample were compared under the same chromatographic condition (HPLC method A) (Figure 1). The results show that each component can be well separated and that related-process impurities can be detected effectively. After purification, the purities of three batches of samples were significantly increased. **SM01**, **SM02-imA**, **SM01-imA**, and **LM49-01-imA** were not detected in the **LM49-01** test sample, but an unknown single impurity **LM49-01-imX** was identified with its content below 0.05%.

In addition, the system of methylene chloride and AlCl_3_ commonly used in Friedel–Crafts acylation was employed in this experiment [15,16]. In consideration of the possibility of chloromethylation, we carefully analyzed the MS and HPLC data of **LM49-01** test sample. No chloromethylation product was observed.

#### 2.2.2. Process Verification of Bromination Step

As shown in Table 9 and Table 10, three batches of **LM49-02** samples were obtained at a 452–490 g scale. In view of the impurities **LM49-02-imA** and **LM49-02-imB** possibly brought from the course of bromination, the control substances of **LM49-01**, **LM49-02-imA**, **LM49-02-imB**, and **LM49-02**, as well as their mixture and **LM49-02** test sample were compared under the same chromatographic condition (HPLC method B) (Figure 2). The results exhibit that each component can be well separated, and related-process impurities can be detected effectively. After purification, the purities of three batches of samples were markedly enhanced. **LM49-01** and **LM49-02-imA** were not detected in the **LM49-02** test sample, only **LM49-02-imB** was identified with its content below 0.42%.

Additionally, in view of the use of NBS in the bromination reaction, the byproduct of NBS is succinimide, which may be a potential impurity in **LM49-02** sample. When the bromination reaction was finished, the mixture was poured into a large excess of ice water. Thereafter, the **LM49-02** crude product was recrystallized with ethanol as the solvent. Because succinimide is soluble in water and ethanol, during this treatment, succinimide can be effectively removed from the reaction mixture. In order to control the limit of impurities, gas chromatography (GC) analysis was applied to test the residue of NBS and succinimide [17]. However, we found NBS was decomposed into succinimide (t_R_ = 8.2 min) during detection. Thus, succinimide as a potential impurity was determined and controlled by GC analysis. After purification, three batches of **LM49-02** samples were analyzed. No succinimide was detected (Figure 3).

#### 2.2.3. Process Verification of Demethylation Step

As presented in Table 11 and Table 12, the final **LM49-API** samples of the three-batch production were obtained at a 329–410 g scale. In consideration of the impurities **LM49-API-imA**, **LM49-API-imB**, and **LM49-API-imC** possibly brought from the course of demethylation, the control substances of **LM49-02**, **LM49-API-imA**, **LM49-API-imB**, **LM49-API-imC**, and **LM49-API** as well as their mixture and **LM49-API** test sample were compared under the same chromatographic condition (HPLC method C) (Figure 4). The results exhibit that each component can be well separated, and related-process impurities can be detected effectively. After purification, the purities of three batches of samples were obviously improved. **LM49-02**, **LM49-API-imA**, and **LM49-API-imB** were not found in the **LM49-API** test sample. The maximal impurity **LM49-API-imC** was identified as the main impurity with its content below 0.03%, respectively. In addition, another unknown impurity **LM49-API-imX** was discovered with its content below 0.02%, its structure was not determined.

### 2.3. Polymorph Investigation of LM49-API

Polymorphs are different solid crystalline forms of the same drug or compound. To the same drug, polymorphs will directly affect its solubility, dissolution, absorption, efficacy, bioavailability, and safety [18,19]. Therefore, the polymorph study is essential to the druggability evaluation. In this paper, different recrystallization conditions were used to inspect the crystal forms of **LM49-API** [20]. Multiple batches of samples were obtained (Table 13), and various analytical methods including differential scanning calorimetry (DSC), thermogravimetry (TG), X-ray powder diffraction (XRD), and infrared spectroscopy (IR) were applied to analyze their crystal forms [21,22,23]. DSC measurement data can analyze the desolvation temperature and melting temperature of the compound. The results in Figure 5A show that the only melting peak (220 °C) of the compound appeared in each batch of samples within the measurement temperature range, which is consistent with the melting point result (219~222 °C), with no solvent endothermic peak found. The TG measurement results in Figure 5B exhibit that each batch of samples displayed two identical weight loss peaks within the temperature range. The compound decomposed quickly after melting, and no obvious solvent weight loss peak was found. XRD results in Figure 5C show that each batch of samples contained the following characteristic peaks expressed in 2θ angles within the measurement angle range: 6.4° ± 0.1°, 9.4° ± 0.1°, 10.1° ± 0.1°, 14.1° ± 0.1°, 18.9° ± 0.1°, and 21.4° ± 0.1°. There was no obvious difference in 2θ angles and peak intensity of the characteristic peaks. The differences in crystal forms lead to different intermolecular forces, modes of action, and intensities of action, and then result in different lattice energies to present different infrared spectra. Further IR characterization was measured. The results in Figure 5D indicate that the infrared absorption spectra of each batch of samples were consistent without obvious peak shift, and all spectra contained the following characteristic absorption peaks: 3482 cm^−1^ (O-H) and 3228 cm^−1^ (=C-H). The above analysis results confirm that the tested **LM49-API** samples did not contain crystal water or other crystal solvents and are of the same crystal form. Moreover, compared with the sample obtained from the original laboratory process, no difference was found.

### 2.4. Impurity Synthesis

During the process development of API, identification of process-related impurities is an important task, which may hamper the API effectiveness and leads to several side effects due to the formation of toxic degradation products [11]. Therefore, process-related impurities must be determined qualitatively and quantitatively. According to the synthetic process and reaction mechanism, we investigated the potential related impurities (shown in Scheme 2), speculated on their formation path, and further prepared them by chemical synthesis methods.

#### 2.4.1. Impurity Analysis

In Friedel–Crafts acylation reaction step, when electrophilic reagent 2-methoxybenzoyl positive ion attacks **SM01**, there are two possible sites at the 3-position and 4-position. When attacking the 4-position, the steric hindrance is smallest, and the resultant product is **LM49-01**. When attacking the 3-position, the by-product **LM49-01-imA** is generated. In addition, **SM01-imA** may be introduced from the purchased raw material **SM01**. Another raw material **SM02** is acyl chloride and is easily hydrolyzed to **SM02-imA**. Therefore, the possible process impurities in this step are **SM01**, **SM01-imA**, **SM02-imA**, and **LM49-01-imA**. However, they are not detected in the test sample of **LM49-01**, but an unknown impurity **LM49-01-imX** is found, whose structure has not been determined yet.

In the bromination reaction step, the main sites of bromine substitution are the 2′- and 5-positions. There may be two mono-bromination by-products, **LM49-02-imA** and **LM49-02-imB**, substituted at the 2′- or 5-positions in the reaction. Therefore, the main process impurities in the bromination reaction are **LM49-01**, **LM49-02-imA**, and **LM49-02-imB**. In fact, only one impurity, **LM49-02-imB**, is detected. Other impurities are not measured in the test samples of **LM49-02** by HPLC. It shows that the 5-position is more active than the 2′-position, and that bromination occurs at the 5-position first. The 2′-position is not conducive to substitution due to the steric hindrance of the adjacent benzene ring.

In a demethylation step, the target product **LM49-API** can be obtained by demethylation of **LM49-02**. Because of the presence of three methoxyl groups in **LM49-02**, there may be the corresponding by-products **LM49-API-imA** and **LM49-API-imB** due to incomplete demethylation. In addition, **LM49-02-imB** exists in multiple batches of **LM49-02** samples, and its demethylation will generate a corresponding impurity **LM49-API-imC**. Therefore, the possible process impurities in demethylation reaction are **LM49-02**, **LM49-API-imA**, **LM49-API-imB**, and **LM49-API-imC**. However, only **LM49-API-imC** and another unknown impurity **LM49-API-imX** are found in the test samples of **LM49-API**.

#### 2.4.2. Preparation of Impurities

In this paper, the possible process-related impurities were prepared by chemical synthesis. In the presence of *n*-butyl lithium, 2-methoxybromobenzene was subjected to halogen–lithium exchange to become 2-methoxyphenyl lithium, and then nucleophilic addition with 2,3-dimethoxybenzaldehyde was carried out to obtain (3,4-dimethoxyphenyl) (2-methoxyphenyl) methanol. Finally, Swern oxidation was conducted to obtain the impurity LM49-01-imA. Under the catalysis of AlCl_3_, 2-methoxybenzoyl chloride and 3,4-dimethoxybromobenzene underwent the Friedel–Crafts acylation reaction to obtain the impurity LM49-02-imA. 5-Bromo 2-methoxybenzoic acid reacted with SOCl_2_ to obtain the corresponding acyl chloride, and then the Friedel–Crafts acylation with 1,2-dimetoxybenzene was carried out to gain LM49-02-imB. During the synthesis of LM49-02, it was found that the mono-brominated impurity LM49-02-imB was also obtained when the molar ratio of LM49-01 to NBS was 1/1. During the demethylation of LM49-02, if the molar ratio of LM49-02 to BBr_3_ was controlled to be 1/1–1/1.5, the impurities LM49-API-imA and LM49-API-imB can be obtained by column chromatography separation. The impurity LM49-02-imB brought from the previous step reacted with BBr_3_ to generate the impurity LM49-API-imC. Finally, more than 5 g of samples with a purity greater than 98% were obtained for all six impurities, which can be used as impurity reference substances for the quality control of LM49-API.

Six prepared impurities were characterized by MS, ^1^H-NMR, ^13^C-NMR, and HRMS. In order to confirm the positions of the hydroxyl groups in **LM49-API-imA** and **LM49-API-imB**, correlation spectra of ^1^H-^1^H COSY, ^1^H-^1^H NOESY, and ^1^H-^13^C HMBC were further employed. The ^1^H-^1^H COSY spectrum data of **LM49-API-imA** show that the low-field aromatic hydrogen signals 7.67 (dd, 1H) and 6.98 (d, 1H) have strong coupling (ortho-position relationship) and belong to 4-Ph-H and 3-Ph-H, respectively. ^1^H-^1^H NOESY spectrum data reveal that the hydrogen signal at 6.98 (d, 1H) is not correlated with high-field methyl hydrogen signals, which indicates that there is no methoxyl group spatially close to it, and that the two methoxyl groups on **LM49-API-imA** should be on another benzene ring. Finally, the hydroxyl group of **LM49-API-imA** was confirmed to be at the 2-position. NOESY spectrum data of **LM49-API-imB** show that 7.22 (s, 1H) is coupled with high-field methyl hydrogen signals, which indicates that this hydrogen atom and the methoxyl group are on the same benzene ring and are spatially close to each other. However, whether the methoxyl group is at the 4′-or 5′-position is still not clear. The ^1^H-NMR data of **LM49-API-imB** indicate that the hydrogen signal 7.22 (s, 1H) may be at the 3′-position or 6′-position. If the position of 7.22 (s, 1H) can be determined, the accurate position of the methoxyl group can be finally confirmed. Further ^1^H-^13^C HMBC results exhibit that the carbonyl carbon signal 195.4 is correlated with the hydrogen signal 6.90 (s, 1H), but is not correlated with the hydrogen signal 7.22 (s, 1H). The hydrogen signals 6.90 (s, 1H) and 7.22 (s, 1H) are therefore attributed to 6′-Ph-H and 3′-Ph-H, respectively. In addition, the hydrogen signal 7.22 (s, 3′-Ph-H, 1H) is found to correlate with the high-field methyl hydrogen signals, which indicates that these hydrogen atoms are adjacent to each other in space. Thus, the methoxyl group in the structure of **LM49-API-imB** can be concluded to belong to the 4′-position.

## 3. Experimental Section

### 3.1. Instruments and Reagents

Melting points were determined on X-4 micro melting point apparatus (Shanghai, China), and were uncorrected. NMR spectra were recorded on a Bruker AVANCE II (Karlsruhe, Germany), 400 MHz or 500 MHz with TMS as an internal standard in DMSO solution. Chemical shifts were given in δ values (ppm), and coupling constants (*J* values) were given in Hz. ESI mass spectra were acquired on a Waters ZQ2000 spectrometer (Milford, MA, USA). HRMS was recorded on a Bruker APEX IV mass spectrometer (Karlsruhe, Germany). IR spectrographs were obtained on the Thermo Scientific Nicolet iS5 (Waltham, MA, USA). XRD was detected on PANalytical X’Pert3 Powder multi-function X-ray diffractometer (Almelo, Dutch). DSC data was obtained from the Differential scanning calorimeter (METTLER TOLEDO, Zurich, Switzerland). TG analysis was detected on a thermogravimetric analyzer (Netzsch, Selb, Germany). HPLC was tested on an Agilent 1260 liquid chromatograph (San Diego, CA, USA) with a DAD detector. HPLC data were obtained using the following methods: HPLC method A: column of Phenomenex-C18 (250 mm × 4.6 mm, 5 µm); mobile phase A (water, adjusted to pH 2.0 with phosphoric acid) and B (acetonitrile). Gradient elution was carried out by adjusting the proportion of mobile phase A/B from 80:20 (20 min) and 60:40 (15 min) to 80:20 (5 min) over 40 min (detection at 210 nm, flow rate of 1.0 mL/min, temperature of 30 °C). HPLC method B: column of Phenomenex-C18 (250 mm × 4.6 mm, 5 µm) and mobile phase A (water, adjusted to pH 2.0 with phosphoric acid) and B (acetonitrile). Gradient elution was carried out by adjusting the proportion of mobile phase A/B from 55:45 (5 min), 48:52 (20 min), 47:53 (5 min), and 46:54 (5 min) to 55:45 (5 min) over 40 min (detection at 230 nm, flow rate of 1.0 mL/min, temperature of 30 °C). HPLC method C: column of Phenomenex-C18 (250 mm × 4.6 mm, 5 µm) and mobile phase A (water, adjusted to pH 2.0 with phosphoric acid) and B (acetonitrile). Gradient elution was carried out by adjusting the proportion of mobile phase A/B from 60:40 (20 min) and 20:80 (3 min) to 60:40 (7 min) over 30 min (detection at 230 nm, flow rate of 1.0 mL/min, temperature of 30 °C). GC data were obtained on an Agilent 7820A Gas Chromatograph (San Diego, CA, USA) with FID detector. GC method: capillary column of Agilent DB-624 (30 m × 0.32 mm, 1.8 μm); split ratio of 10:1; temperature programming: initial column temperature of 120 °C for 0 min with a 20 °C/min ramp to 160 °C, held at 160 °C for 3 min; next with a 20 °C/min ramp to 220 °C, and held at 220 °C for 12 min; carrier gas for N_2_; flow rate of 1.0 mL/min; injection temperature of 240 °C, detector temperature of 250 °C.

The main reagents including **SM01** and **SM02** were provided from Aikon International Ltd. (Nanjing, China). Their purities were controlled to be higher than 99%, the content of maximal single impurity was limited to be less than 0.5%. NBS was purchased from Shanghai Hanhong Chemical, Co., Ltd. (Shanghai, China). DBDMH was obtained from Shanghai Aladdin Bio-Chem Technology Co., Ltd. (Shanghai, China). PyBr_3_, BBr_3_, and AlCl_3_ were purchased from Beijing MREDA Technology Co., Ltd. (Beijing, China). Succinimide was provided by Shanghai Macklin Biochemical Co., Ltd. (Shanghai, China). Other chemical reagents and solvents were commercially available unless otherwise indicated.

### 3.2. Improved Prepared Process of LM49-API

As can be seen in Scheme 1, **LM49-API** was prepared starting from **SM01** and **SM02** by Friedel–Crafts acylation, bromination, and demethylation reaction.

#### 3.2.1. Preparation of Intermediate 2,4′5′-Trimethoxybenzophenone (LM49-01)

Anhydrous aluminum chloride (289 g, 2.17 mol) was added to 3.0 L CH_2_Cl_2_ and kept at a temperature of 15 °C with stirring. **SM01** (300 g, 2.17 mol) was slowly added into the mixture, which was gradually changed to a light-yellow transparent solution. Then **SM02** (409 g, 2.28 mol) was slowly dropped into the system, which changed the color from light yellow to red-brown and allowed the solution to warm to room temperature. The reaction process was monitored by TLC. After being stirred for 1 h at room temperature, the mixture was slowly poured into 12 L ice water. The organic phase was separated, and the water phase was extracted once with 3.0 L CH_2_Cl_2_. The combined organics were washed with 3.0 L 5% NaOH water solution, dried over anhydrous sodium sulfate, and then concentrated via rotary evaporation to obtain the light-yellow oily liquid, which was next dissolved in 0.6 L CH_2_Cl_2_. 12 L *n*-hexane was added, stirred, and kept at 20 °C for 8 h. The precipitation was filtered and dried at 35 °C to gain the white solid (461 g in 76.7% yield with 99.97% HPLC purity). Mp 59–60 °C; ^1^H-NMR (400 MHz, DMSO-*d*_6_) δ 7.51 (td, *J* = 1.8 Hz, 1H), 7.39 (d, *J* = 2.0 Hz, 1H), 7.26 (dd, *J* = 7.4, 1.8 Hz, 1H), 7.17 (d, *J* = 1.8, 1H), 7.15 (dd, *J* = 15.6, 2.0 Hz, 1H), 7.07 (td, *J* = 7.5 Hz, 1H), 7.03 (d, *J* = 8.5 Hz, 1H), 3.84 (s, 3H), 3.80 (s, 3H), 3.70 (s, 3H); ^13^C-NMR (100MHz, DMSO-*d*_6_) δ 194.6, 156.8, 153.8, 149.2, 131.8, 130.3, 129.3, 128.8, 125.8, 120.8, 112.2, 111.1, 110.9, 56.2, 55.9, 55.9; MS (ESI) *m*/*z* (100%) 273 [M + H]^+^.

#### 3.2.2. Preparation of Key Intermediate 5,2′-Dibromo-2,4′5′-Trimethoxybenzophenone (LM49-02)

NBS (1137 g, 6.39 mol) was added into the solution of **LM49-01** (435 g, 1.60 mol) in 2.2 L THF. The mixture was stirred for 5 h at room temperature, then poured into 8 L ice water to quench the reaction. The precipitation was filtered and separated to get the yellow-like solid, which was recrystallized at a proportion of 1 g/10 mL of **LM49-02**/ethanol and dried at 50 °C to afford 460 g white granular solid in 83.0% yield with 99.82% HPLC purity. Mp 133–134 °C; ^1^H-NMR (400 MHz, DMSO-*d*_6_) δ 7.74 (dd, *J* = 8.9, 2.5 Hz, 1H), 7.54 (d, *J* = 2.5 Hz, 1H), 7.21 (s, 1H), 7.13 (d, *J* = 8.9 Hz, 1H), 7.03 (s, 1H), 3.74 (s, 3H), 3.66 (s, 3H), 3.30 (s, 3H); ^13^C-NMR (100MHz, DMSO-*d*_6_) δ 192.8, 157.9, 151.8, 148.3, 136.5, 132.8, 132.5, 130.1, 116.6, 115.5, 113.7, 112.4, 111.6, 56.7, 56.6, 56.3; MS (ESI) *m*/*z* (100%) 429,431,433 [M + H]^+^.

#### 3.2.3. Preparation of the Target Compound (LM49-API)

**LM49-02** (450 g, 1.05 mol) was added to 2.25 L CH_2_Cl_2_, stirred to dissolve completely. Next, BBr_3_ (786 g, 3.14 mol) was slowly dropped into the system at –15 °C. Then the mixture was allowed to warm to room temperature. After stirring for 2 h, the reaction mixture was slowly poured into 9 L ice water to quench the reaction. The precipitation was filtered to obtain the yellow solid, which was recrystallized with the mixed solvent of ethanol and water with the 1 g/4 mL/4 mL proportion of **LM49-API**/ethanol/water and dried at 50 °C in a vacuum oven to afford 350 g light yellow crystal solid in 88.3% yield with 99.98% HPLC purity. Mp 220–221 °C; ^1^H-NMR (400 MHz, DMSO-*d*_6_) δ 10.83 (s, 1H), 10.10 (brs, 1H), 9.61 (brs, 1H), 7.62 (dd, *J* = 8.8, 2.6 Hz, 1H), 7.39 (d, *J* = 2.6 Hz, 1H), 7.02 (s, 1H), 6.95 (d, *J* = 8.8 Hz, 1H), 6.89 (s, 1H); ^13^C-NMR (100MHz, DMSO-*d*_6_) δ 195.4, 157.8, 149.1, 144.6, 137.0, 133.2, 129.4, 125.1, 119.8, 119.6, 117.4, 109.8, 108.3; MS (ESI) *m/z* (100%)386, 389, 391 [M + H]^+^.

### 3.3. Polymorphs Investigation of LM49-API

The target product **LM49-API** was recrystallized with different single or mixed solvents to obtain different batches of samples, which were further analyzed by DSC, TG, XRD, and IR techniques.

DSC determination: nitrogen protection, flow rate 50.0 mL min^−1^, heating rate 10.0 K min^−1^, temperature range 25–250 °C. TG measurement: argon gas protection, heating rate 5.0 K min^−1^, temperature range 25–1020 °C. XRD determination: tube pressure 40 kV, tube flow 40 mA, Cu Target Kα radiation, array detector detection, scan step 0.013°, scanning speed 19° min^−1^, 2θ scanning range 3–40°, continuous scanning mode, room temperature (25 °C). IR detection: scanning range 4000–400 cm^−1^, resolution 4 cm^−1^, scan times 5.

### 3.4. Preparation of Process-Related Impurities

#### 3.4.1. 2,2′3′-Trimethoxylbenzophenone (LM49-01-imA)

2-Bromoanisole (25 g, 13.4 mmol) was dissolved in 100 mL anhydrous THF kept at a temperature of −78 °C with protection of nitrogen. The solution of *n*-butyllithium (*n*-BuLi) in hexane (2.5 mol·L^−1^, 60 mL) was slowly added into the system. After stirring for 30 min, 2,3-dimethoxybenzaldehyde (22 g, 13.4 mol) was dissolved in the 50 mL anhydrous THF, which was slowly dropped into the mixture. The reaction mixture was stirred for 4 h, then poured into 200 mL water to quench reaction, which was twice extracted with 400 mL ethyl acetate. The combined organics phase was dried by anhydrous sodium sulfate and concentrated via rotary evaporation to obtain 34.6 g light yellow solid. Crude product was purified by silica gel column chromatography with the system of petroleum ether-ethyl acetate (*v*/*v*, 6/1) as the eluent to afford 26.3 g white solid-compound **LM49-01-imA-01**.

Oxalyl chloride 6.15 mL was added to 100 mL CH_2_Cl_2_ protected by nitrogen. Then the mixed solvent of 16.5 mL DMSO and 35 mL CH_2_Cl_2_ was dropped into the reaction mixture slowly at −78 °C. After stirring for 1 h, the solution of 10 g **LM49-01-imA-01** in 15 mL DMSO was slowly dropped into the system. After stirring for 2 h, the temperature of the mixture was allowed to warm to 0 °C, and 50 mL triethylamine was added dropwise into the system. After stirring for 30 min, the reaction mixture was poured into 300 mL of water. Then the separated organic phase was washed with 100 mL saturated sodium chloride solution and dried over anhydrous sodium sulfate. The dried organic phase was concentrated via rotary evaporation to obtain the off-white solid. The crude product was further purified by silica gel column chromatography with the system of petroleum ether-ethyl acetate (*v*/*v*, 3/1) as the eluent to afford 9.8 g white solid-compound **LM49-01-imA** in 70.8% yield with 99.16% HPLC purity (Scheme 3). Mp 97–98 °C; ^1^H-NMR (400 MHz, DMSO-*d*_6_) δ 7.52 (m, 1H), 7.38 (d, *J* = 7.6 Hz, 1H), 7.21 (d, *J* = 8.2 Hz, 1H), 7.12 (m, 2H), 7.03 (m, 1H), 6.93 (d, *J* = 7.8 Hz, 1H), 3.83 (s, 3H), 3.62 (s, 3H), 3.43 (s, 3H); ^13^C-NMR (100MHz, DMSO-*d*_6_) δ 195.4, 158.1, 152.8, 147.4, 135.5, 133.4, 130.1, 130.0, 124.2, 120.7, 120.7, 116.1, 112.6, 60.9, 56.4, 56.0; MS (ESI) *m/z* (100%) 273.2 [M + H]^+^.

#### 3.4.2. 2′-Bromo-2,4′,5′-Trimethoxylbenzophenone (LM49-02-imA)

Anhydrous aluminum chloride (3.7 g, 28.0 mmol) was added to 50 mL CH_2_Cl_2_. Then, 3,4-dimethoxybromobenzene (6.6 g, 28.0 mmol) and 2-methoxybenzoyl chloride (5 g, 36.0 mmol) were slowly added into the mixture at 0 °C, respectively. The temperature of mixture was allowed to warm to room temperature, stirred for 2 h, then poured into 200 mL water to quench the reaction. The organic phase was separated, and the aqueous phase was extracted twice with 50 mL CH_2_Cl_2_. The combined organics were dried over anhydrous sodium sulfate and concentrated via rotary evaporation to get light yellow oil liquid. The crude product was purified by silica gel column chromatography with the system of petroleum ether-ethyl acetate (*v*/*v*, 6/1) as the eluent to give 7.0 g white solid-compound **LM49-02-imA** in 66.0% yield with 100% HPLC purity (Scheme 4). Mp 73–74 °C; ^1^H-NMR (400 MHz, DMSO-*d*_6_) δ 7.61–7.53 (m, 1H), 7.44 (dd, *J* = 7.6, 1.8 Hz, 1H), 7.19 (s, 1H), 7.14 (d, *J* = 8.4 Hz, 1H), 7.05 (td, *J* = 1.0 Hz, 1H), 6.97 (s, 1H), 3.85 (s, 3H), 3.71 (s, 3H), 3.66 (s, 3H); ^13^C-NMR (100MHz, DMSO-*d*_6_) δ 194.2, 158.8, 151.4, 148.3, 134.5, 133.6, 131.2, 128.0, 121.0, 116.5, 113.4, 113.0, 111.1, 56.6, 56.3, 56.2; MS (ESI) *m/z* (100%) 351.1, 353.1 [M + H]^+^; HRMS (ESI) calculated for C_16_H_16_BrO_4_ [M + H]^+^ 351.0232, found 351.0222.

#### 3.4.3. 5-Bromo-2,3′,4′-Trimethoxylbenzophenone (LM49-02-imB)

5-Bromo-2-methoxybenzoic acid (5 g, 21.6 mmol) was dissolved in 25 mL anhydrous SOCl_2_, which was heated to reflux for 4 h in the presence of DMF and then concentrated under reduced pressure to afford 5-bromo-2-methoxybenzoyl chloride. Then 50 mL of anhydrous CH_2_Cl_2_ was added into the system, and anhydrous AlCl_3_ (2.88 g, 21.6 mmol) and 1,2-dimethoxybenzene (2.99 g, 21.6 mmol) were added to the system at 0 °C, respectively. The reaction mixture was allowed to warm to room temperature, stirred for 2 h, and then slowly poured into 100 mL of water to quench the reaction. The organic phase was separated, and the aqueous phase was extracted twice with 50 mL CH_2_Cl_2_. The combined organics were dried over anhydrous sodium sulfate, concentrated under reduced pressure, and purified by silica gel column chromatography with the system of methanol-CH_2_Cl_2_ (*v*/*v*, 1/80) as the eluent to obtain 5 g white solid-compound **LM49-02-imB** in 66.0% yield with 99.19% HPLC purity (Scheme 5). Mp 114–115 °C; ^1^H-NMR (500 MHz, DMSO-*d*_6_) δ 7.68 (dd, *J* = 8.9, 2.5 Hz, 1H), 7.43 (d, *J* = 2.5 Hz, 1H), 7.39 (d, *J* = 1.9 Hz, 1H), 7.17–7.14 (m, 2H), 7.03 (d, *J* = 8.5 Hz, 1H), 3.84 (s, 3H), 3.81 (s, 3H), 3.70 (s, 3H); ^13^C-NMR (100MHz, DMSO-*d*_6_) δ 192. 8, 156.1, 154.1, 149.2, 134.2, 131.3, 131.0, 129.7, 126.0, 114.8, 112.3, 111.3, 110.8, 56.4, 56.3, 56.0; MS (ESI) *m/z* (100%) 351.1, 353.0, 354.3 [M + H]^+^; HRMS (ESI) calculated for C_16_H_16_BrO_4_ [M + H]^+^ 351.0232, found 351.0212.

#### 3.4.4. 5,2′-Dibromo-2-Hydroxyl-4′,5′-Dimethoxylbenzophenone (LM49-API-imA)

**LM49-02** (7 g, 16.3 mmol) was dissolved in 50 mL CH_2_Cl_2_, and 4.1 g (16.3 mmol) BBr_3_ was slowly dropped into the system at 0 °C. The reaction mixture was allowed to warm to room temperature, stirred for 1 h, and poured into 150 mL ice water to quench the reaction. The organic phase was separated, and the water phase was extracted with 50 mL CH_2_Cl_2_. The combined organic phase was dried on anhydrous sodium sulfate and concentrated under reduced pressure to give the crude product, which was further purified by silica gel column chromatography with the system of petroleum ether-ethyl acetate-dichloromethane (*v*/*v*/*v*, 6/1/1) as the eluent to acquire 5 g light yellow solid-compound **LM49-API-imA** in 74.1% yield with 98.72% HPLC purity (Scheme 6). Mp 130–131 °C; ^1^H-NMR (400 MHz, DMSO-*d*_6_) δ 11.03 (s, 1H), 7.67 (dd, *J* = 8.8, 2.6 Hz, 1H), 7.41 (d, *J* = 2.6 Hz, 1H), 7.25 (s, 1H), 7.11 (s, 1H), 6.98 (d, *J* = 8.8 Hz, 1H), 3.86 (s, 3H), 3.75 (s, 3H); ^13^C-NMR (100MHz, DMSO-*d*_6_) δ 196.9 159.3, 151.4, 148.5, 138.5, 134.3, 131.9, 124.3, 120.5, 116.2, 113.0, 110.5, 110.2, 56.6, 56.4; MS (ESI) *m/z* (100%)415.3, 416.2 [M + H]^+^; HRMS (ESI) calculated for C_15_H_12_Br_2_O_4_ [M − H]^−^ 414.9004, found 414.9002.

#### 3.4.5. 5,2′-Dibromo-2,5′-Dihydroxyl-4′-Methoxylbenzophenone (LM49-API-imB)

**LM49-02** (20 g, 46.5 mmol) was dissolved in 100 mL CH_2_Cl_2_, and 17.5 g BBr_3_ (69.7 mmol) was added to the system slowly at 0 °C. The mixture was allowed to warm to room temperature, stirred for 1 h, and poured into 400 mL water to quench the reaction. The precipitation was removed by filtration. The separated organic phase was dried over anhydrous sodium sulfate and then concentrated under reduced pressure to obtain the crude product, which was purified by silica gel column chromatography with the system of petroleum ether-ethyl acetate-dichloromethane (*v*/*v*/*v*, 6/1/1) as the eluent to obtain 5.1g yellow solid-compound **LM49-API-imB** in 27.3% yield with 99.40% HPLC purity (Scheme 7). Mp 198–199 °C; ^1^H-NMR (400 MHz, DMSO-*d*_6_) δ 10.90 (s, 1H), 9.68 (s, 1H), 7.64 (dd, *J* = 8.8, 2.6 Hz, 1H), 7.39 (d, *J* = 2.6 Hz, 1H), 7.21 (s, 1H), 6.95 (d, *J* = 8.8 Hz, 1H), 6.90 (s, 1H), 3.86 (s, 3H); ^13^C-NMR (100MHz, DMSO-*d*_6_) δ 196.3, 158.7, 150.8, 146.3, 138.0, 133.9, 131.7, 125.1, 120.3, 117.0, 116.9, 110.4, 108.5, 56.6; HRMS (ESI) calculated for C_14_H_10_Br_2_O_4_ [M + H]^+^ 402.9004, found 402.9822.

#### 3.4.6. 5-Bromo-2,3′,4′-Trihydroxylbenzophenone (LM49-API-imC)

**LM49-02-imB** (6 g, 17.1 mmol) was dissolved in 50 mL CH_2_Cl_2_. BBr_3_ (17.2 g, 68.4 mmol) was dropped into the system at 0 °C. The mixture was allowed to warm to room temperature, stirred for 1 h, and poured into 200 mL ice water slowly. The precipitate was filtered and dried at 50 °C in a vacuum oven to obtain 5.2 g yellow solid-compound **LM49-API-imC** in 98.5% yield with 99.52% HPLC purity (Scheme 8). Mp 112–113 °C; ^1^H-NMR (400 MHz, DMSO-*d*_6_) δ 10.18 (s, 1H), 9.92 (brs, 1H), 9.40 (brs, 1H), 7.49 (dd, *J* = 8.7, 2.6 Hz, 1H), 7.33 (d, *J* = 2.6 Hz, 1H), 7.19 (d, *J* = 2.1 Hz, 1H), 7.08 (dd, *J* = 8.3, 2.1 Hz, 1H), 6.90 (d, *J* = 8.7 Hz, 1H), 6.82 (d, *J* = 8.3 Hz, 1H); ^13^C-NMR (100 MHz, DMSO-*d*_6_) δ 193.7, 155.0, 151.6, 145.6, 134.2, 131.4, 129.6, 128.9, 123.6, 119.0, 116.9, 115.7, 110.3; MS (ESI) *m/z* (100%) 309.0, 311.0 [M + H]^+^; HRMS (ESI) calculated for C_13_H_9_BrO_4_ [M − H]^−^ 306.9606, found 306.9604.

## 4. Conclusions

The improved **LM49-API** preparation process addresses the main issues encountered in the original laboratory synthesis by changing the reacted reagents and optimizing the conditions. In particular, the key intermediate **LM49-02** is facilely prepared using NBS as a bromination reagent and a simple post-treatment method. Ultimately, these improvements lead to an efficient, economic, and easy preparation of **LM49-API** in three steps, with mild reaction conditions, low energy consumption, high purity, and controllable impurities. The preparation scale of **LM49-API** attains to 329–410 g, with the HPLC purity greater than 99.95%, total impurities less than 0.05%, and maximum single impurity less than 0.03%, with no polymorphic phenomenon found. Meanwhile, six potential process-related impurities were also prepared to obtain the samples of over 5 g with the HPLC purity higher than 98%, respectively, which can be used as the corresponding impurity reference substances. Through HPLC analysis, the main impurities in **LM49-API** contain **LM49-API-imC** and **LM49-API-imX**, the content of which is less than 0.03%.

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
