# Peer review of "Process Development and Synthesis of Process-Related Impurities of an Efficient Scale-Up Preparation of 5,2′-Dibromo-2,4′,5′-Trihydroxy Diphenylmethanone as a New Acute Pyelonephritis Candidate Drug"

_molecules, 2020, doi:10.3390/molecules25030468_

Round 1

Reviewer 1 Report

The authors illustrate the study and optimization of the synthesis of 5,2′-3 Dibromo-2,4′,5′-trihydroxy diphenylmethanone in large scale. This compound showed interesting activity against acute Pyelonephritis and, in order to perform the preclinical studies, a revisited scale-up preparation process suitable for commercial production was required. Moreover, a polymorphism analysis and investigation of some impurities have been conducted.

The work appears well organized and the compounds properly characterized. Anyway, some parts need improvements. So, I recommend the current report for publication after a revision of the manuscript in order to clarify some aspects.

The following comments have to be considered:

Scheme 1, pag 3: the authors compare two syntheses of compound LM49-API. It should be useful to have the two synthetic routes in the scheme in order to make the differences more evident for readers. Line 89, pag 3: the authors affirm to have improved the first step avoiding the formation of the acyl chloride, that requires the use of SOCl2, although the acyl chloride is the starting material. They have to explain how they got it, comparing the difference costs of the starting materials. Table 3: the time of the process must be added in the note. Line 149, pag 5: mono-brominated LM49-02-imB is mentioned. Please, add a scheme or figure with the structures of the molecules described in the text. Table 4: The solvent used (ethanol) must be explicated in the note. Line 339, pag 16: I suggest to move schemes 2,3,4 from experimental part to this paragraph. In this way, the reading will be easier. Line 428, pag 18: please, add experimental details for recrystallization with ethanol (amount of solvent, temperature…)

Reviewer 2 Report

The manuscript describes the development of a large scale synthesis of a pharmaceutical agent. Activities such as this are commonplace in the drug discovery and development sphere to prepare sufficient quantities, for example, for toxicological studies.

Overall, I think the authors have done themselves a huge injustice by very poor presentation of their research. There are far too many language errors to mention and the authors need to have the manuscript either double checked by a native English speaker or use an editing service. Some errors are subtle (antibiotics are "misused" and not "abused", line 36), some errors are probably due to picking the incorrect word ("ammonia water " is in fact aqueous ammonia and methanol is a class 2 solvent and not a "second class" solvent) and some make it difficult to understand what authors mean (sentence in line 43-45).

There are also a number of other issues that in rewriting the manuscript, authors should fix. Use of low temp or neat bromine, or having exothermic reaction are not particularly prohibitive on the scale the authors are using (less than a kilograms). Having to deal with the succinimide byproduct however would be (i'll come to that).

I must say, much of the detailed developmental work carried out by the authors doesn't seem to me that novel, in the sense that they appear to be obvious, or that interesting to a reader. For example using different crystal purities when using different solvent mixtures, isn't surprising, or informative to a reader. Overall, inclusion of this much detail, gives the impression that the work in the paper is routine rather than novel.

Similarly, the final section of the paper starts by describing the importance of polymorphism, only to show there is no polymorphism. So again, it does not read well.

For these reasons, I cannot support publication of the manuscript in its current format. I strongly recommend the authors use either an editing service, or ask an English speaker to read and fix the manuscript. The authors should also rewrite the paper with summerising the "obvious" experimental details. 

From a scientific content, I think the authors need to say something about purity (and consistency of the purity) of the 2-methoxybenzoylchloride. Also, the authors should look at the procedure for bromination.  the byproduct of NBS is succinimide, but it is not clear to me how that is removed from reaction mixture. The use of methylene chloride and AlCl3 has an obvious drawback that it can result in chloromethylation (also a Friedel-Crofts) and the authors should state that it was not observed. Finally, the authors must give details of previously reported characterisation of all compounds, it they are previously reported.
